# Hypothermic Oxygenated Machine Perfusion (HOPE) Prior to Liver Transplantation Mitigates Post-Reperfusion Syndrome and Perioperative Electrolyte Shifts

**DOI:** 10.3390/jcm11247381

**Published:** 2022-12-12

**Authors:** Fabian Horné, Moritz Drefs, Malte Joachim Schirren, Dominik Thomas Koch, Ganildo Cepele, Severin Johannes Jacobi, Elnaz Payani, Nikolaus Börner, Jens Werner, Markus Otto Guba, Dionysios Koliogiannis

**Affiliations:** Department of General, Visceral and Transplant Surgery, University Hospital of Munich, 81377 Munich, Germany

**Keywords:** post-reperfusion syndrome, PRS, HOPE, hypothermic oxygenated machine perfusion, liver transplantation, organ preservation, machine perfusion

## Abstract

(1) Background: Post-reperfusion syndrome (PRS) and electrolyte shifts (ES) represent considerable challenges during liver transplantation (LT) being associated with significant morbidity. We aimed to investigate the impact of hypothermic oxygenated machine perfusion (HOPE) on PRS and ES in LT. (2) Methods: In this retrospective study, we compared intraoperative parameters of 100 LTs, with 50 HOPE preconditioned liver grafts and 50 grafts stored in static cold storage (SCS). During reperfusion phase, prospectively registered serum parameters and vasopressor administration were analyzed. (3) Results: Twelve percent of patients developed PRS in the HOPE cohort vs. 42% in the SCS group (*p* = 0.0013). Total vasopressor demand in the first hour after reperfusion was lower after HOPE pretreatment, with reduced usage of norepinephrine (−26%; *p* = 0.122) and significant reduction of epinephrine consumption (−52%; *p* = 0.018). Serum potassium concentration dropped by a mean of 14.1% in transplantations after HOPE, compared to a slight decrease of 1% (*p* < 0.001) after SCS. The overall incidence of early allograft dysfunction (EAD) was reduced by 44% in the HOPE group (*p* = 0.04). (4) Conclusions: Pre-transplant graft preconditioning with HOPE results in higher hemodynamic stability during reperfusion and lower incidence of PRS and EAD. HOPE has the potential to mitigate ES by preventing hyperpotassemic complications that need to be addressed in LT with HOPE-pre-treated grafts.

## 1. Introduction

Liver transplantation (LT) is the only therapeutic option for patients with end-stage liver disease and certain liver malignancies in a curative setting. While its relevance in curing fatal diseases is undisputed, LT is also prone to several complications that might not only prolong convalescence but also jeopardize graft and, hence, recipient survival. Among these complications, one is particularly feared by LT physicians as it adversely affects further organ function immediately right at the beginning of the actual transplant process—post-reperfusion syndrome (PRS).

PRS in liver transplantation was first described by Aggarwal et al. in 1987 and defined as a 30% decrease in mean arterial pressure (from baseline), lasting for at least 1 min and occurring within 5 min after vessel unclamping in graft reperfusion [1]. Incidence rates vary considerably across studies, ranging from 12 to 77% [2]. Risk factors include recipient age, MELD-score, high creatinine, low hemoglobin and low calcium levels, as well as cardiac parameters such as left-ventricular diastolic dysfunction, elevated heart rate and low central venous pressure [3,4,5,6,7,8,9,10]. Typically, PRS is associated with a decrease in heart rate and systemic vascular resistance and an increase in central venous pressure and pulmonary capillary wedge pressure. However, the severity and duration of PRS are unpredictable and have a strong impact on patient morbidity and mortality [5,7,11]. Occurrence of PRS increases the need for blood transfusions [3,9,12], prolongs ICU and hospital length of stay [3,12] and negatively affects renal function after transplantation [7,11]. Several studies provide an overview and further classification of PRS, with a review by Manning et al., summarizing the differing definitions and describing risk factors for the development of PRS and its potential long-term implications [13].

In addition to the hemodynamic effects during reperfusion, electrolyte imbalances can occur, with hyperpotassemia being the most feared. The rapid influx of potassium is thought to originate from the preservation fluid remaining in the graft at the time of reperfusion. The resulting systemic hyperpotassemia can occur in varying degrees of severity and can lead to cardiac arrhythmias and even cardiac arrest. PRS seems to be associated with the development of intraoperative hyperpotassemia [6]. To counteract these effects, mechanisms have been developed to ensure flushing or venting of the graft. Different solutions and routes of flushing (e.g., portalvenous, arterial, or hepatic venous) have been established and their effects have been described in many studies. However, these studies yielded conflicting conclusions in terms of electrolyte changes and short- and long-term clinical outcomes [14,15,16].

The origin of most clinically relevant complications—that might also derive from or result in PRS—is the so-called ischemia-reperfusion injury (IRI) that occurs when oxygen-enriched blood hits ischemic cells. The consecutive formation of reactive oxygen species (ROS) amongst other cellular pathways mediates cell damage [17,18]. One of the notable clinical milestones in preventing excessive IRI in liver grafts was the clinical introduction of oxygenated machine perfusion in 2010 [19]. Since then, several positive effects on patients’ short- and long-term outcomes have been reported. The possibilities offered by machine perfusion are manifold. Most notably, hypothermic oxygenated machine perfusion (HOPE) has been implemented in many LT centers worldwide in recent years, primarily due to its simple initiation process. Essentially, HOPE protects liver grafts from initial reperfusion injury, leading to better graft function and fewer biliary complications compared to SCS-treated organs, to name a few advantages [20,21].

In our center, HOPE treatment of liver grafts prior to transplantation was first introduced in 2019 and has been established as a standard of care prior to every liver transplantation since 2020. Over time, we have been able to confirm the positive effects as described in the literature with regard to excellent post-transplant outcomes. While previous studies focused mainly on mid- and long-term effects of machine perfusion, the present study was conducted to shed light on the immediate perioperative phase and to analyze differences in donor organ reperfusion after previous HOPE or SCS treatment in terms of hemodynamic parameters, vasopressor requirements, and electrolyte shifts—thus assessing the impact of HOPE on PRS.

## 2. Materials and Methods

### 2.1. Study Design

This study was designed as a retrospective analysis of perioperative variables in patients undergoing liver transplantation who were treated in a single reference center for hepatobiliary surgery and liver transplantation. The study was approved by the Institutional Review Board of the LMU University of Munich (protocol number EK-LMU 19-395).

### 2.2. Hypothermic Oxygenated Machine Perfusion (HOPE)

In 2019, HOPE using the LiverAssist^®^ device (XVIVO, Groningen, The Netherlands, and Göteborg, Sweden) was introduced in our transplant department and was established as a standard of care for liver graft preconditioning in 2020. Starting in November 2020, all liver grafts that underwent pre-transplant hypothermic machine perfusion were included in our analysis. All liver grafts were procured after declaration of brain death, preserved and transported in HTK solution on ice. Upon arrival, grafts were prepared for implantation and then connected to the LiverAssist device by cannulation of the portal vein. Organs were perfused with a University of Wisconsin machine perfusion solution (UW-MPS) at 8–12 °C and a portal vein pressure of 3–5 mmHg. Perfusion was continued while the recipient underwent hepatectomy. Grafts were subsequently flushed with HTK solution and implanted.

### 2.3. Static Cold Storage (SCS)

The control group consisted of transplantations performed before implementation of HOPE as a standard of care. These grafts were stored on ice in HTK solution throughout back table preparation and during the recipient’s hepatectomy until the implant phase of the transplantation was reached.

### 2.4. Perioperative Data Sources

Induction of anesthesia, monitoring, and invasive line placement was carried out in standard fashion. Dissection and surgical technique for transplantation were not altered in or influenced by this study. During surgery, mean arterial blood pressure was monitored by invasive catheterization. Potassium levels were measured using sequential blood gas analyses. Administration of vasopressors was tracked at 60 min after skin incision—as a baseline value—and then 60 and 5 min prior to, as well as 30 s, 5 and 60 min after reperfusion. Infusion of fluids and transfusion of blood products was recorded prospectively.

### 2.5. Outcomes

The primary outcome was the incidence of PRS during the reperfusion phase of a liver transplantation—defined according to literature as a 30% decrease in mean arterial pressure (from baseline), lasting for at least 1 min and occurring within 5 min after vessel unclamping in graft reperfusion [1]. Secondary outcomes included: (1) effect on hemodynamics; (2) consecutive dose-correlated use of vasopressors; (3) imbalances in serum electrolytes—especially potassium levels. The incidence of early allograft dysfunction (EAD) was assessed using criteria defined by Olthoff et al., including the presence of one or more of the following previously defined postoperative laboratory analyses reflective of liver injury and function: bilirubin > or = 10 mg/dL on day 7, international normalized ratio > or = 1.6 on day 7, and alanine or aspartate aminotransferases >2000 IU/L within the first 7 days [22].

### 2.6. Donor-Specific Variables

Donor-specific variables included the following parameters: Age, sex, Eurotransplant Donor Risk Index (DRI), cold ischemic time, as well as warm ischemic time. These variables were obtained from the Eurotransplant database.

### 2.7. Recipient-Specific Variables

Recipient-specific variables included the following parameters: Age, sex, body mass index (BMI), match-MELD score, American Association of Anesthesiologists (ASA) Score, CHILD-Pugh Score, healthcare setting prior to transplant, number and type of comorbidities, as well as indication for transplant. These variables were retrieved from our liver surgery database and supplemented with data from electronic patient records.

### 2.8. Statistical Analysis

Univariate comparisons were performed calculating Fisher’s exact test, Chi-Square test and Mann–Whitney U Test for non-parametrical comparisons. All predictors and variables were included as time-constant variables, using values from the assessment at the time of liver transplantation. For statistical analysis, SPSS statistical software package (version 25, IBM, Chicago, IL, USA) and GraphPad Prism (version 8.4.2, GraphPad Software, San Diego, CA, USA) were used. *p*-value (two-tailed) of <0.05 is regarded as statistically significant.

## 3. Results

We included 100 patients in this study—50 of whom received HOPE-treated liver grafts in 2020 and 2021. They were compared to 50 patients who received an SCS-treated graft in 2018 and 2019 before our institution started regular HOPE utilization.

### 3.1. Donor Characteristics

Baseline donor characteristics are shown in Table 1. No relevant difference in donor specific variables were registered, as both groups were very similar in terms of age distribution, sex, Eurotransplant DRI, cold ischemic time and warm ischemic time.

### 3.2. Recipient Characteristics

Baseline recipient characteristics are shown in Table 2. Both groups were very similar in terms of age distribution, body mass index (BMI), match-MELD score and 30-day survival. In the SCS group, the proportion of male patients was more pronounced, however without statistical significance. Indications for LT were distributed similarly amongst both collectives. It is worth noting that the HOPE group carried a significantly higher disease burden in terms of ASA Score (*p* = 0.013), CHILD Score, number of relevant comorbidities and the need for high-level care in intensive or intermediate care units prior to transplantation, thus showing a slight tendency towards more morbid patients. Within the HOPE cohort, a mean perfusion time of 163.8 min [95% Confidence Interval (CI): 138.3–189.3] was registered.

### 3.3. Incidence of Post-Reperfusion Syndrome

In the cohorts studied, the incidence of post-reperfusion syndrome, as defined by Aggarwal et al., was distinctly lower in the perfused liver grafts. While only 12% of cases (*n* = 6 out of 50) in the HOPE group met the criteria for PRS, 42% (*n* = 21 out of 50) in the SCS group developed post-reperfusion syndrome. This difference was also highly significant (*p* = 0.0013). Respective graphic analysis is shown in Figure 1A.

### 3.4. Hemodynamic Chracteristics

Perioperative hemodynamic variables are displayed in Table 3. After reperfusion, mean immediate drop in mean arterial pressure (MAP) was 12.7% [95% CI: 8.8–16.5] in the HOPE group, compared to 28.2% [95% CI: 24.1–32.4] in the SCS group (*p* < 0.001). As shown in Figure 1B, HOPE preconditioned livers showed a significantly mitigated decrease in MAP compared to SCS-transplants. This also led to a relevantly lower need for vasopressors (Figure 1C,D): While patients in the HOPE group required an increased administration of norepinephrine of a median of 750 µg (95% CI: 614–1041) in the first hour after reperfusion, SCS controls required 960 µg (95% CI: 830–1405). Median increase in epinephrine administration was 96 µg in the HOPE-treated patients (95% CI: −368–355), whereas an increased administered amount of median 199 µg [95% CI: 212–412] was registered in SCS patients. Thus, HOPE preconditioning resulted in a −22% (*p* = 0.122) reduction in the amount of norepinephrine administered and a −56% (*p* = 0.0184) significantly lower requirement for epinephrine compared to SCS. There were no relevant differences between the groups in vasopressin administration (*p* = 0.899).

### 3.5. Serum Characteristics and Electrolyte Balance

Blood potassium levels dropped by a mean of 14.1% [95% CI: 10.0–18.1] in livers after HOPE, requiring intravenous substitution in 40% of cases, whereas potassium levels showed no relevant decrease (−1% [95%CI: −3.4–5.5]; *p* < 0.001) in SCS grafts, requiring substitution in only 16% of patients (*p* = 0.0135) (Table 3 and Figure 2). Thus, potassium levels decreased to a significantly lower baseline with a significantly higher need for substitution after HOPE pretreatment, as compared to SCS grafts. Notably, a 28.8% higher transfusion requirement of fresh frozen plasma (FFP) was observed in the HOPE cohort compared to the SCS group (*p* = 0.045). Apart from this, no significant differences in fluid or blood product substitution were observed between the two groups (Table 3).

### 3.6. Incidence of Early Allograft Dysfunction and Primary Non-Function: Thirty-Day Survival

In the cohorts studied, the incidence of EAD was distinctly lower in the perfused liver grafts. While only 28% of cases (*n* = 14 out of 50) in the HOPE group met the criteria for EAD, 50% (*n* = 25 out of 50) in the SCS group developed EAD within the first seven days post-transplant (Table 2 and Figure 3; *p* = 0.04). In addition, no occurrence of PNF could be observed in the HOPE group compared with three cases in the SCS control cohort; no significant difference in 30-day survival was found in the studied groups (Table 2).

## 4. Discussion

It is fair to say that the implementation of organ machine perfusion represents a crucial innovation in LT. Over the past decade, an increasing number of authors have shared their clinical and experimental findings on machine perfusion—almost invariably with positive results on post-transplant graft function, patient and graft survival and graft acceptance rates [20,23,24,25,26,27]. However, the choice of the “best” machine perfusion modality remains controversial—with the majority of healthcare professionals opting for HOPE or normothermic machine perfusion (NMP) [27]. In the case of NMP, well-described viability criteria provide crucial information on the graft’s condition in a pseudo-physiological setting prior to transplantation [28,29]. This advantage of NMP over HOPE provides additional safety in graft selection, resulting in a reduction in discard rates [25]. In addition, NMP offers the theoretical possibility for drug-associated graft pre-treatment [30]. On the other hand, most favorable evidence on NMP is available for grafts from donors with circulatory death (DCD). Since only grafts from brain-dead donors (DBD) are approved for transplantation in Germany, our department chose HOPE as the standard preconditioning procedure for liver grafts.

Many studies share the unanimous opinion that HOPE should be considered the “better” graft storage technique compared to SCS, providing true graft pre-treatment prior to transplant [23,26,27]. A vast number of benefits of HOPE have been well documented to date and are primarily reflected in a reduction of IRI, better post-transplant graft function and fewer biliary complications [20,21,23,31]. Furthermore, there is growing evidence that, even in HOPE, viability testing is possible [27,32]. In addition to these well-known advantages of HOPE over SCS, our study demonstrates more beneficial effects of HOPE, occurring during graft reperfusion, affecting short-term patient outcomes and presumably long-term outcomes as well.

The results of our study indicate a significant difference in the occurrence of PRS during LT, with criteria for PRS being met in only six cases in the HOPE group, compared to 21 cases in the SCS group. An additional remarkable observation made was that hemodynamic instabilities were less frequent in the HOPE group and, when they did occur, better manageable with lower amounts of vasopressors. Prudent use of a vasopressor (or a combination of 2) with targeted fluid bolus administration and adjustments in anesthetic levels may have a profound effect on MAP. One proposed mechanism for this MAP drop is myocardial stunning and transient loss of contractility due to an abrupt increase in serum potassium, decreased pH, and decreased temperature of the reperfusion blood that subsequently enters the systemic circulation [33]. In addition to myocardial stunning, reperfusion is known to trigger the release of vasodilators that can further reduce MAP [34,35].

Furthermore, inflammatory agents are thought to play a role in the pathophysiology of the reperfusion syndrome, with a correlation to increased complement cascade activation [36] and immune response [37]. A proposed mechanism accounting for the advantages of HOPE over SCS may lie in the elimination of the vasodilatory and inflammatory agents, and a subsequent reduction of potassium concentration during graft perfusion. An additional factor contributing to potassium depletion after reperfusion might be ATP availability in the graft [38]. In standard cold storage, cell metabolism and oxygen consumption are reduced by hypothermia. Nonetheless, anaerobic processes continue at a low rate, eventually leading to ATP depletion [39] and accumulation of ischemic signature metabolites such as succinate [40]. As ionic stability is largely attributed to ATP-dependent Na+/K+ carriers, intracellular depletion can lead to deregulation, calcium accumulation and acidification [41]. These detrimental effects that occur during ischemia contribute largely to the development of IRI and are aggravated by prolongation of cold ischemic time [42]. HOPE, in contrast, uses the benefits of reduced metabolism during hypothermia, yet facilitates unlimited electron flow along the respiratory chain within the mitochondrium, resulting in reduced ROS formation and elevated ATP synthesis [26]. As hypothermic oxygenated perfusion has been shown to replenish ATP-levels and reduce succinate concentrations [43], we hypothesize that, following reperfusion, operating ion carriers lead to an uptake of potassium into the graft, hence depleting serum levels.

While this is considered beneficial in terms of hemodynamic stability, overshooting effects have been observed. The resulting hypopotassemia may itself pose dangers and has made potassium substitution necessary in a substantial number of cases. One possible explanation could be that hyperpotassemia after graft-reperfusion is expected by anesthesiologists [44]. This is mainly due to organ preservation and carrying solutions that are high in potassium and not cleared in a standardized manner in earlier years [44]. In practice, this might lead to a liberal use of insulin–glucose solutions in order to shift potassium into the cells. The impact of hyperpotassemia on the occurrence of PRS itself remains controversial [6,45]. Meanwhile, hypopotassemia has been previously described as a complication after HOPE [21], and this finding is supported by the results of our study. With ex-vivo perfusion becoming more common, these observations need to be highlighted so that anesthesiologists can anticipate electrolyte shifts and adapt their medical treatment accordingly.

In fact, there are some limitations to our study: the most obvious one is the retrospective nature of it. Nevertheless, we believe that, due to the significance and clear tendencies of our gained results, this study manages to provide relevant insights into one of the numerous beneficial effects of HOPE preconditioning prior to LT. Several studies have already assessed the effects of machine perfusion on perioperative results and postoperative outcomes; however, most of them have only demonstrated positive effects on PRS and EAD in cohorts with DCD grafts. In fact, the protective effects are even more pronounced in DCD-only cohorts and have been observed after both HOPE and NMP. While both techniques appear to shorten the length of hospital stay in recipients, HOPE is less prone to procedure-related complications [46]. Then, again, NMP allows viability testing under physiological conditions and reduces the length of the ICU stay [25,46]. However, very limited data are available for NMP in DBD-only cohorts with limited power in terms of outcome and none in terms of PRS [46]. Current literature suggests that machine perfusion exerts its benefits mainly in grafts with increased marginality. The influence of machine perfusion on less marginal organs is therefore still debated: Study groups from Zurich (CH) and Aachen (DE) recently conducted randomized controlled trials (NCT01317342 and NCT03124641) with HOPE-pretreated DBD grafts compared to SCS. While the Aachen group only included grafts from donors with extended criteria, the Zurich group allowed quite broad inclusion criteria in their study, comparable to the cohort of recipients at our department in this study [47]. However, the focus of both studies was on postoperative outcome data and not on perioperative findings, thus differing significantly from our study. While the Aachen group was already able to demonstrate the positive effects of HOPE preconditioning in ECD-DBD grafts, the transplant research community is waiting with great interest for the results of the Zurich study, especially in DBD-only countries [48].

The selection and composition of the two groups investigated is another limitation to our study. Since HOPE replaced SCS as the standard of care procedure for graft preconditioning prior to LT at our center, we were not able to select different LT cohorts transplanted at approximately the same time points. However, in order to minimize the risk of selection bias, performance bias and variation in standard practices in the performance of liver transplantation over time, we chose to include patients transplanted at closely spaced time points, directly prior and directly after implementing HOPE preconditioning at our institution. The retrospective study design and its observational approach imply a heterogenous group distribution and, although the sample size was not small, the present differences in cohort composition need to be addressed. Overall, LT recipients or waiting list candidates per se represent a rather heterogenous cohort of patients [49]. While the studied groups are very similar in terms of age, BMI and match-MELD-Score, the HOPE population appeared to carry a higher disease burden prior to transplantation. For instance, the mean ASA score was significantly higher in the HOPE group (3.72 vs. 3.46; *p* = 0.013) and more patients required treatment in intensive or intermediate care wards (30% vs. 18%; *p* = 0.16) prior to transplantation, compared to SCS controls. Respecting these observations, the favorable effects of HOPE on the development of PRS described in this study might as well be interpreted as even more valuable than previously thought. Hence, one could conclude that, even in a sicker patient population, the hemodynamic benefits achieved by HOPE are still significantly beneficial. Clearly, further prospective evaluation is needed to support the findings of this study.

## 5. Conclusions

Pre-transplant graft perfusion through HOPE provides a number of benefits. In this study, we demonstrate an impact on both patient hemodynamics and post-reperfusion syndrome. HOPE results in greater hemodynamic stability during reperfusion with a reduced need for vasopressors and lower incidence of post-reperfusion syndrome as compared to SCS-treated grafts. Furthermore, we were able to demonstrate effects in electrolyte balance. In particular, a decrease in potassium levels after reperfusion was observed in the HOPE group, which in many cases even had to be compensated. The risk for hyperpotassemic cardiac arrhythmias—as described in previous years—can thus be significantly reduced. Whereas in the past a preventive potassium shift by glucose/insulin administration was often required, this should be omitted in HOPE-perfused livers to avoid hypopotassemia. Instead, possible potassium substitution should be anticipated.

## Figures and Tables

**Figure 1 jcm-11-07381-f001:**
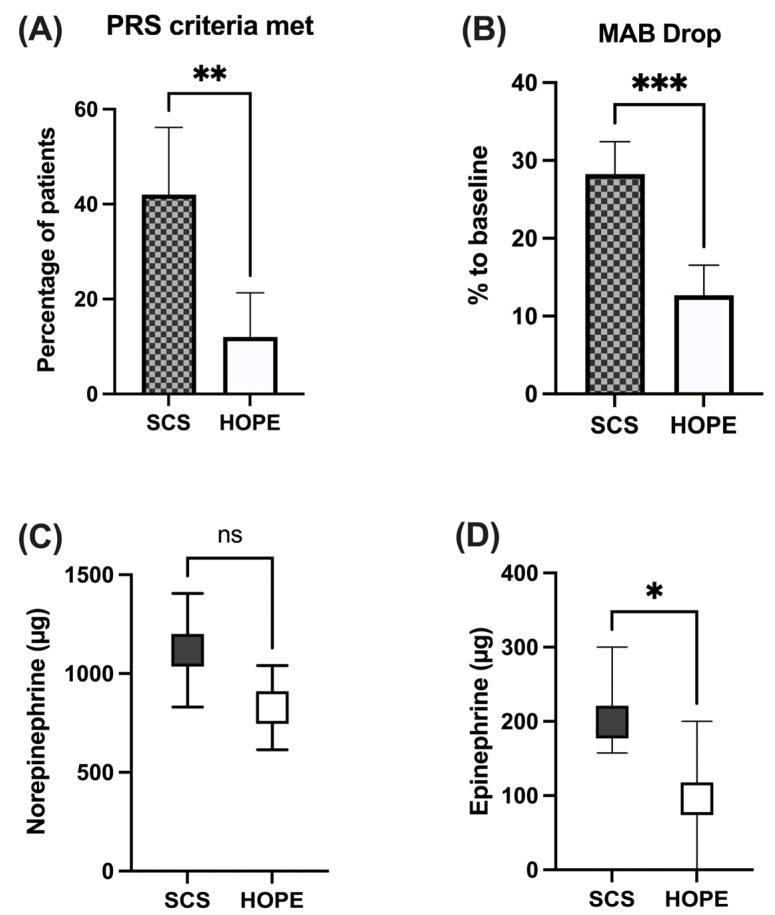
Graphical analysis of perioperative findings in the immediate post-reperfusion phase. (**A**) ratio of post reperfusion syndrome (PRS) occurrence in liver graft recipients (Mean ± 95% CI). HOPE preconditioning results in significantly reduced incidence of PRS as compared to SCS (*p* = 0.0013); (**B**) relative decrease in mean arterial blood pressure (MAP) immediately after liver graft reperfusion (Mean ± 95% CI). HOPE preconditioning leads to significantly reduced drop in MAP as compared to SCS (*p* < 0.0001); (**C**,**D**) absolute increase in vasopressor demand within the first hour after liver graft reperfusion, as compared to the hour prior to reperfusion (Median ± 95%-CI). Compared to SCS, HOPE preconditioning significantly reduces administered doses of epinephrine (**D**) *p* = 0.018) and shows a trend towards reduced norepinephrine doses (**C**) *p* = 0.122). (* = *p* < 0.05, ** = *p* < 0.01, *** = *p* < 0.001, ns = not significant).

**Figure 2 jcm-11-07381-f002:**
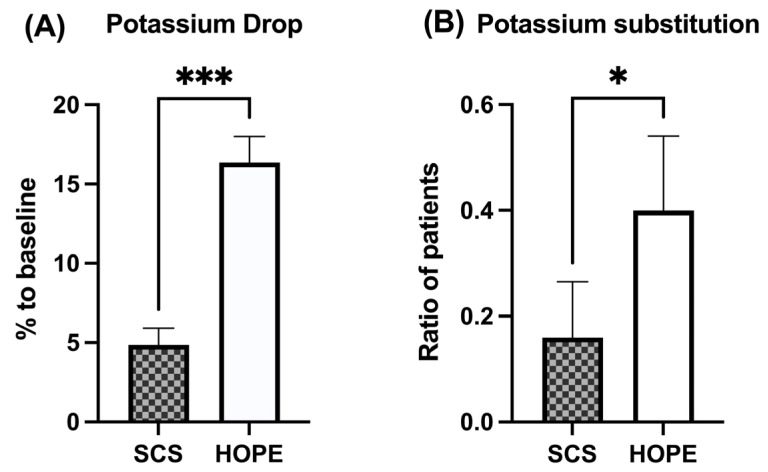
Graphical analysis of relative decrease in serum potassium levels and need for potassium substitution, immediately after liver graft reperfusion (Mean ± 95%-CI). HOPE preconditioning results in significantly higher drops in serum potassium levels (**A**) *p* < 0.001) and a more frequent need for potassium substitution (**B**) *p* = 0.014), as compared to SCS. (* = *p* < 0.05, *** = *p* < 0.001).

**Figure 3 jcm-11-07381-f003:**
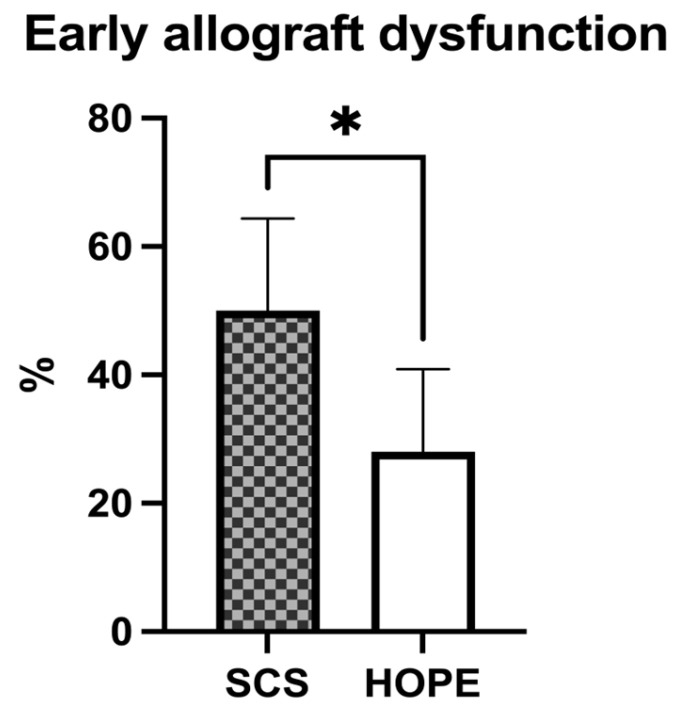
Graphical analysis of early allograft dysfunction in liver transplant recipients (Mean ± 95%-CI). HOPE preconditioning results in significantly lower incidence of EAD within the first seven postoperative days, as compared to SCS. (* = *p* < 0.05).

**Table 1 jcm-11-07381-t001:** Description of characteristics of liver graft donors and ischemic times of the respective grafts, stratified by liver graft preservation technique.

Donor Characteristics and Ischemic Times at Liver Transplantation	All, N (%)/Mean [95% CI]	Liver Graft Preservation Techniques		*p*-Value (HOPE vs. SCS)
		-SCS- N (%)/Mean [95% CI]	-HOPE- N (%)/Mean [95% CI]	
Sex				0.69
Male	51 (51.0%)	24 (48.0%)	27 (54.0%)	
Female	49 (49.0%)	26 (52.0%)	23 (46.0%)	
Age (y)				
	52.1 [48.7–55.4]	55.1 [50.5–59.6]	49.2 [44.4–54.0]	0.05
Donor Risk Index (DRI)				
	2.16 [2.08–2.24]	2.21 [2.08–2.33]	2.11 [2.00–2.33]	0.38
Cold Ischemic Time (CIT)				
	603.9 [574.0–633.7]	599.0 [554.3–643.7]	608.0 [566.5–649.6]	0.96
Warm Ischemic Time (WIT)				
	55.3 [51.1–59.6]	52.4 [46.4–58.5]	58.1 [52.0–64.2]	0.25
Total	100 (100%)	50 (50.0%)	50 (50.0%)	

CI: Confidence Interval; CIT: Cold Ischemic Time; DRI: Donor Risk Index; HOPE: Hypothermic Oxygenated Perfusion; SCS: Static Cold Storage; WIT: Warm Ischemic Time; y: years.

**Table 2 jcm-11-07381-t002:** Description of characteristics and key variables of liver transplant recipients stratified by liver graft preservation technique.

Recipient Characteristics and Control Variables at Date of Liver Transplantation	All, N (%)/Mean [95% CI]	Liver Graft Preservation Techniques		*p*-Value (HOPE vs. SCS)
		-SCS- N (%)/Mean [95% CI]	-HOPE- N (%)/Mean [95% CI]	
**Sex**				0.086
Male	68 (68.0%)	38 (76.0%)	30 (60.0%)	
Female	32 (32.0%)	12 (24.0%)	20 (40.0%)	
**Age (y)**				
	51.8 [49.7–54.0]	50.6 [47.2–54.0]	53.0 [50.3–55.8]	0.37
**BMI**				
	25.4 [24.5–26.3]	25.4 [24.1–26.8]	25.3 [24.0–26.7]	0.80
**match-MELD Score**				
	21.3 [19.3–23.3]	20.4 [17.6–23.2]	22.1 [19.2–25.1]	0.44
**ASA Score**				
	3.6 [3.5–3.7]	3.5 [3.3–3.6]	3.7 [3.6–3.9]	**0.013**
**Healthcare setting prior to transplant**				0.16
ICU/IMC	24 (24.0%)	9 (18.0%)	15 (30.0%)	
Regular ward/outpatient	76 (76.0%)	41 (82.0%)	35 (70.0%)	
**CHILD-Pugh Score**				0.43
No cirrhosis	20 (20.0%)	12 (24.0%)	8 (16.0%)	
A	20 (20.0%)	12 (24.0%)	8 (16.0%)	
B	31 (31.0%)	14 (28.0%)	17 (34.0%)	
C	29 (29.0%)	12 (24.0%)	17 (34.0%)	
**Number of Comorbidities ***				0.32
0–1	55 (55.0%)	30 (60.0%)	25 (50.0%)	
2–3	35 (35.0%)	14 (28.0%)	21 (42.0%)	
≥4	10 (10.0%)	6 (12.0%)	4 (8.0%)	
Mean	1.7 [1.4–1.9]	1.6 [1.2–2.1]	1.7 [1.3–2.0]	0.52
**Indication for transplant**				0.42
Acute Liver Failure	14 (14.0%)	9 (18.0%)	5 (10.0%)	
Alcoholic Cirrhosis	29 (29.0%)	11 (22.0%)	18 (36.0%)	
HCC	25 (25.0%)	16 (32.0%)	9 (18.0%)	
NASH	3 (3.0%)	1 (2.0%)	2 (4.0%)	
PSC	19 (19.0%)	10 (20.0%)	9 (18.0%)	
Viral Hepatitis	24 (24.0%)	16 (32.0%)	8 (16.0%)	
Autoimmune Hepatitis	15 (15.0%)	8 (8.0%)	7 (14.0%)	
Other **	21 (21.0%)	10 (20.0%)	11 (22.0%)	
**Early allograft dysfunction (EAD)**				**0.04**
EAD	39 (39.0%)	25 (50.0%)	14 (28.0%)	
No EAD	61 (61.0%)	25 (50.0%)	36 (72.0%)	
**Primary Non Function (PNF)**				0.24
PNF	3 (3.0%)	3 (3.0%)	0 (0.0%)	
No PNF	97 (97.0%)	47 (47.0%)	50 (50.0%)	
**30-day Survival**				0.36
Survival	96 (96.0%)	47 (94%)	49 (98%)	
No Survival	4 (4.0%)	3 (6.0%)	1 (2.0%)	
**Total**	100 (100%)	50 (50.0%)	50 (50.0%)	

ASA: American Society of Anesthesiologists; BMI: Body Mass Index; CI: Confidence Interval; EAD: Early Allograft Dysfunction; HCC: Hepatocellular Carcinoma; HOPE: Hypothermic Oxygenated Perfusion; ICU: Intensive Care Unit; IMC: Intermediate Care Unit; MELD: Mean of End-Stage Liver Disease; min: minutes; NASH: Non-alcoholic Steatosis Hepatis; PSC: Primary Sclerosing Cholangitis; SCS: Static Cold Storage. * Further explanation of relevant comorbidities is demonstrated in Appendix A. ** Further explanation of other transplant indications is demonstrated in Appendix A. Bold *p*-values < 0.05.

**Table 3 jcm-11-07381-t003:** Description of perioperative hemodynamic variables and fluid/transfusion management during liver transplantation, stratified by the liver graft preservation technique.

Perioperative Variables and Fluid/Transfusion Management during Liver Transplantation	All, N (%)/Mean [95% CI]	Liver Graft Preservation Techniques		*p*-Value (HOPE vs. SCS)
		-SCS- N (%)/Mean [95% CI]	-HOPE- N (%)/Mean [95% CI]	
**Packed RBC (mL)**				
	1315 [1049–1521]	1235 [846–1625]	1395 [1021–1769]	0.40
**FFP (mL)**				
	2487 [2094–2880]	2174 [1616–2732]	2800 [2242–3358]	**0.045**
**Platelet Transfusion (mL)**				
	366 [261–471]	294 [159–429]	438 [274–602]	0.19
**Machine Auto-Transfusion (mL)**				
	663 [465–863]	603 [377–829]	724 [389–1059]	0.60
**Crystalloid Infusion (mL)**				
	6006 [5470–6542]	6161 [5517–6805]	5851 [4973–6730]	0.09
**Colloid Infusion (mL)**				
	529 [433–624]	602 [458–746]	455 [328–582]	0.09
**Tranexamic acid (mg)**				
	601 [429–773]	731 [447–1016]	471 [273–669]	0.26
**MAB Drop (%)**				
	20.5 [17.3–23.7]	28.2 [24.1–32.4]	12.7 [8.8–16.5]	**<0.001**
**PRS Criteria**				**0.0013**
PRS Criteria met	27 (27.0%)	21 (42.0%)	6 (12.0%)	
PRS Criteria not met	73 (73.0%)	29 (58.0%)	44 (88.0%)	
**Increase in Norepinephrine (µg)**				
	973 [795–1151]	960 [830–1405]	750 [614–1041]	0.12
**Increase in Epinephrine (µg)**				
	153 [−34–340]	199 [212–412]	96 [−368–355]	**0.018**
**Increase in Vasopressin (IE)**				
	0.31 [0.17–0.45]	0.29 [0.10–0.49]	0.33 [0.11–0.54]	0.899
**Potassium Drop (%)**				
	7.6 [4.3–10.8]	−1 [−3.4–5.5]	14.1 [10.0–18.1]	**<0.001**
**Potassium Substitution**				**0.0135**
Substitution	28 (28.0%)	8 (16.0%)	20 (40.0%)	
No Substitution	72 (72.0%)	42 (84.0%)	30 (60.0%)	
**Total**	100 (100%)	50 (50.0%)	50 (50.0%)	

CI: Confidence Interval; FFP: Fresh Frozen Plasma; HOPE: Hypothermic Organ Perfusion; mg: milligrams; mL: milliliters; PRS: Post-Reperfusion syndrome; RBC: Red Blood Cells; SCS: Static Cold Storage. Bold *p*-values < 0.05.

## Data Availability

The data presented in this study are available on request from the corresponding author. The data are not publicly available due to ethical and privacy reasons.

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
