# Peer review of "Hypothermic Oxygenated Machine Perfusion (HOPE) Prior to Liver Transplantation Mitigates Post-Reperfusion Syndrome and Perioperative Electrolyte Shifts"

_jcm, 2022, doi:10.3390/jcm11247381_

Round 1
Reviewer 1 Report
In the present study the authors compare the intraoperative hemodynamic parameters and the electrolyte shifts in 50 LT with HOPE vs 50 LT with SCS. Although the inherent limitations of the retrospective nature of the study it is well conducted and give an important information that has been missed up to now as it is the intraoperative period of LT when perfusion machine is used. I congratutated the authors for that.
I suggest to include in the discussion a brief comparison of the results with those coming DBD o DCD recipients with NMP
In order to increase the impact of the paper, it would be nice that the authors show the intraoperative blood product transfusion as well as the tranexanic acid administration. Also, they could explain the intraoperative fluid policy and coagulation management to better interpretate the data.
Finaly, one year or at least one month postransplant survival should be added
Reviewer 2 Report
“In a retrospective study design and its observational approach, the use of hypothermic oxygenated machine perfusion for liver transplantation reduced the incidence of post- reperfusion syndrome. This was also proven from the hemodynamic point of view.”
This is a very interesting paper to me, but I have a few questions.
①How many bases were used for each of the 50 cases compared and how were they extracted ? Propensity matching score analysis was not possible?
â‘¡ How did donor characteristics age, gender, and donor risk index compare?
â‘¢ Figure.1, 2, and 4 would be better for Table.
â‘£ If this study was conducted in the immediate perioperative phase, it may be necessary to evaluate early allograft dysfunction.
⑤ Vasopressin in Fig. 3 C is not used in both groups and there is no difference. Is it necessary to present it?
â‘¥ The discussion is detailed, but also includes more than was gleaned from the data in this study. It should be more concise.
⑦ Is there any comment on the ongoing clinical trials (NCT01317342, NCT03124641) to see if perfusion with a hypothermia device is effective in liver transplantation from brain-dead donors?
Round 2
Reviewer 2 Report
It has been carefully revised in response to comments. The discussion from references 45,48 could provide a new explanation for the effectiveness of HOPE. We also believe that the paper is good throughout.
Author Response
Thank you for your efforts and for reviewing our work again!